# MiR-23a Regulates Skin Langerhans Cell Phagocytosis and Inflammation-Induced Langerhans Cell Repopulation

**DOI:** 10.3390/biology12070925

**Published:** 2023-06-28

**Authors:** Jie Wang, Nirmal Parajuli, Qiyan Wang, Namir Khalasawi, Hongmei Peng, Jun Zhang, Congcong Yin, Qing-Sheng Mi, Li Zhou

**Affiliations:** 1Center for Cutaneous Biology and Immunology Research, Department of Dermatology, Henry Ford Health, Detroit, MI 48202, USA; jwang5@hfhs.org (J.W.); nparaju1@hfhs.org (N.P.); qwang1@hfhs.org (Q.W.); cyin1@hfhs.org (C.Y.); 2Immunology Research Program, Henry Ford Cancer Institute, Henry Ford Health, Detroit, MI 48202, USA; 3Department of Medicine, College of Human Medicine, Michigan State University, East Lansing, MI 48824, USA; 4Department of Biochemistry, Microbiology and Immunology, School of Medicine, Wayne State University, Detroit, MI 48202, USA; 5Department of Internal Medicine, Henry Ford Health, Detroit, MI 48202, USA

**Keywords:** miR-23a, Langerhans cells, skin

## Abstract

**Simple Summary:**

Controlling epigenetic signals for mediating the homeostatic immune response in skin-residing Langerhans cells (LCs) ontogeny, maintenance, and function are largely not determined. Here, we demonstrated that the epigenetic regulator miR-23a not only plays an indispensable role for LC antigen uptake but is also crucial for bone marrow-derived LC repopulation during the inflammatory condition. During this process, miR-23a may regulate LC phagocytosis by mediating efferocytosis and endocytosis-mediated targets as well as LCs repopulation via targeting Fas and Bcl-2 family proapoptotic molecules. Instead, miR-23a did not demonstrate playing any role in LC development and postnatal homeostasis. Thus, the context-dependent regulatory role of miR-23a in LCs may represent an extra-epigenetic layer for incorporating TGF-β and PU.1-mediated regulation during steady-state and inflammation-induced repopulation.

**Abstract:**

Langerhans cells (LCs) are skin-resident macrophage that act similarly to dendritic cells for controlling adaptive immunity and immune tolerance in the skin, and they are key players in the development of numerous skin diseases. While TGF-β and related downstream signaling pathways are known to control numerous aspects of LC biology, little is known about the epigenetic signals that coordinate cell signaling during LC ontogeny, maintenance, and function. Our previous studies in a total miRNA deletion mouse model showed that miRNAs are critically involved in embryonic LC development and postnatal LC homeostasis; however, the specific miRNA(s) that regulate LCs remain unknown. miR-23a is the first member of the miR-23a-27a-24-2 cluster, a direct downstream target of PU.1 and TGF-b, which regulate the determination of myeloid versus lymphoid fates. Therefore, we used a myeloid-specific miR-23a deletion mouse model to explore whether and how miR-23a affects LC ontogeny and function in the skin. We observed the indispensable role of miR-23a in LC antigen uptake and inflammation-induced LC epidermal repopulation; however, embryonic LC development and postnatal homeostasis were not affected by cells lacking miR23a. Our results suggest that miR-23a controls LC phagocytosis by targeting molecules that regulate efferocytosis and endocytosis, whereas miR-23a promotes homeostasis in bone marrow-derived LCs that repopulate the skin after inflammatory insult by targeting Fas and Bcl-2 family proapoptotic molecules. Collectively, the context-dependent regulatory role of miR-23a in LCs represents an extra-epigenetic layer that incorporates TGF-b- and PU.1-mediated regulation during steady-state and inflammation-induced repopulation.

## 1. Introduction

Langerhans cells (LCs) are skin-resident macrophages that act similarly to dendritic cells (DCs) for controlling adaptive immunity and immune tolerance in the skin. LCs constitute 1–2% of epidermal cells and characteristically express C-type lectin langerin [1]. Unlike bone marrow (BM)-derived conventional DCs, LCs originate from embryonic yolk sac macrophages and fetal liver monocytes [2,3]. After birth, adult LCs are maintained at an appropriate density through self-renewal under steady-state conditions, but under inflammatory conditions, monocyte-derived LC precursors repopulate the skin [4,5]. As professional antigen presenting cells, immature epidermal LCs constantly capture and process external and internal antigens, migrating to nearby draining lymph nodes where they complete their maturation process and present antigens to naïve T cells. These mature LCs also secrete cytokines and provide costimulatory signals that influence whether the T cells promote an immunogenic or tolerogenic response [6]. As potent immunoregulatory cells, LCs participate in the pathogenesis of various skin disorders, including infections, allergy, autoimmune diseases, and cancer [7,8,9]. Several studies, including ours, have shown that the embryonic development, postnatal differentiation, maintenance, and functional activation of LCs are tightly controlled by dynamically regulated transcriptional programs, including TGF-β and related downstream signaling pathways [10]. However, detailed insights on how specific epigenetic factors, such as microRNAs (miRNAs), connect and coordinate different transcriptional programs and signaling pathways to regulate LC activity are still lacking. 

MicroRNAs (miRNAs), a class of small non-coding RNAs, regulate protein-coding genes by inhibiting mRNA translation. Our previous studies, which used a conditional Dicer deletion mouse model, have revealed the critical role of miRNAs in the embryonic development of LCs [11] and dendritic epidermal T cells [12]. The critical involvement of miRNAs in postnatal LC homeostasis and function has also been demonstrated in a DC-specific Dicer deletion mouse model [13]. However, the individual miRNAs involved in LC regulation and related molecular mechanisms remain largely unknown. miR-23a is the first member of the miR-23a-27a-24-2 cluster, which is well-conserved among various species [14]. The miR-23a-27a-24-2 cluster has been implicated in multiple physiological and pathological processes, including hematopoietic progenitor cell production and differentiation [15,16], osteoblast differentiation [17,18], and effector T cell differentiation and expansion [19,20], and it has been reported as both an oncogene and tumor suppressor gene involved in tumor development [21]. More interestingly, TGF-β and PU.1, which are the essential regulators of LC development and function, also promote miR-23a expression [22,23], suggesting a potential role for miR-23a in TGF-β signaling during LC regulation. Hence, we hypothesized that miR-23a plays a role in LC development, homeostasis, and functional regulation. Using a myeloid lineage-specific Csf1riCre-mediated miR-23a deletion mouse model, we uncovered an indispensable role for miR-23a in LC antigen uptake and inflammation-induced LC epidermal repopulation, even though the embryonic development and postnatal homeostasis of LCs were not affected when miR-32a was lacking. RNA-sequencing (RNA-seq) revealed distinctive target molecules and signaling pathways involved in the miR-23a-mediated regulation of LCs during the steady state and inflammation-induced LC repopulation. 

## 2. Materials and Methods 

### 2.1. Mice

Csf1r^iCre^ mice (#021024) were purchased from Jackson Laboratory. The miR-23a^fl/fl^ mice were kindly provided by Dr. Qi-Jing Li, Duke University [20]. All mice were backcrossed to C57BL/6J mice for at least six generations. Experiments were conducted on 7- to 12-week-old mice that were age and gender matched, unless otherwise indicated. The mice were bred and maintained in the pathogen-free animal facility in the Department of Bioresources at Henry Ford Health. The handling of mice and experimental protocols were in accordance with the Institutional Animal Care and Use Committee of Henry Ford Health.

### 2.2. Epidermal Single-Cell Suspension Preparation

Mouse skin was harvested and processed as previously described [24]. Briefly, trunk skin was first incubated in 0.5% Dispase (Gibco, Billings, MT, USA) for 1 h at 37 °C. Then, epidermal sheet pieces were digested in complete culture medium containing 0.01% DNase (Sigma, St Louis, MO, USA) for 1 h at 37 °C. Samples were filtered through a 40 µm mesh, and single epidermal cells were collected for flow cytometry or sorting.

### 2.3. Flow Cytometry and Antibodies

Single-cell suspensions were pretreated with anti-FcγRII/III (clone 2.4G2) for 10 min at 4 °C, and then stained for surface and intracellular markers with the following conjugated monoclonal antibodies: I-A/E (MHC-II) (M5/114.15.2), CD45 (30-F11), CD207 (4C7), EpCAM (G8.8), F4/80 (BM8), CD11c (N418), CD11b (M1/70), CD80 (16-10A1), CD86 (GL1), Ly6G (1AB-Ly6g), CD115 (c-fms), Ly6C (HK1.4), and CD172a (SIRPα). All antibodies were purchased from eBioscience/Invitrogen (San Diego, CA, USA), Tonbo Biosciences (San Diego, CA, USA), or BioLegend (San Diego, CA, USA). All flow cytometry data were collected using BD FACS Celesta (BD Biosciences, San Jose, CA, USA); LCs were sorted using BD FACS Aria II (BD Biosciences, San Jose, CA, USA); and data were analyzed with FlowJo software version 10.7.1 (TreeStar, Sam Carlos, CA, USA). P0 LC, adult LC, and peripheral blood mononuclear cell (PBMC) gating strategies are shown in Appendix A.

### 2.4. LC In Vitro Antigen Uptake Assay

Freshly isolated epidermal cells were incubated with 0.025% dextran-fluorescein isothiocyanate (FITC) (Life Technologies, Grand Island, NY, USA) for 45 min at 37 °C or 4 °C. Then, cells were washed with phosphate-buffered saline (PBS) and subsequently stained with anti-CD45 and anti-MHC-II antibodies. The percentage of LCs that took up antigen (CD45+ MHC-II+ FITC+) was determined via flow cytometry.

### 2.5. LC Replenishment during Skin Inflammation

To deplete LCs, mice were exposed to ultraviolet irradiation (UV) for 15 min (wavelength 254 nm, voltage 8 W, source: 38 cm). LC network replenishment was assessed 3 weeks after UV treatment [24,25].

### 2.6. In Vitro LC Maturation

Freshly separated epidermal cells were cultured in complete culture medium at 37 °C for 72 h. Cells were then collected and stained with anti-MHC-II, CD207a, CD45, CD80, and CD86 antibodies for flow cytometry.

### 2.7. RNA Extraction and qRT-PCR 

MicoRNA was isolated using miRNeasy Tissue/Cells Advanced Mini Kit (Qiagen), and RNA was extracted using the GenEluteTM Total RNA Purification Kit (Sigma). qRT-PCR was performed using the QuantStudio 7 Flex Real-Time PCR System. Primer sequences used for miR-23a (R: 5′-GCGGAGTGTAGCACCTAAAGAGAG-3′ and F: 5′-CCTAGACTCCAGAGGTAGAGGCAG-3′) as well as those for other target genes, namely ABCA1 (F: 5′-AGTTTCGGTATGGCGGGTTT-3′ and R: 5′-AGCATGCCAGCCCTTGTTAT-3′), MAP4K4 (F: 5′-CCAGACGCCACTTACGACTAC-3′ and R: 5′-CGCTGCATGTAATTCTTCACCA-3′), VEGF-a (F: 5′-GGAGATCCTTCGAGGAGCACTT-3′ and R: 5′-GGCGATTTAGCAGCAGATATAAGAA-3′), NF1 (F: 5′-TCAAGCATGGACTTGGCACT-3′ and R: 5′-CATTCGTATTGCTGGGTGCG-3′), and GAPDH (F: 5′-GGTGAAGGTCGGTGTGAACG-3′, R: 5′-TGTAGACCATGTAGTTGAGGTCA-3′) were quantified in a real-time PCR reaction using a Universal SYBR Green Master mix (#S4320 ROX, Roche, Indianapolis, IN, USA). Relative expression was calculated using ^ΔΔ^C_T_ method.

### 2.8. Bulk RNA Sequencing

RNA concentration was determined using the Qubit RNA broad range assay in the Qubit Fluorometer (Invitrogen, Waltham, MA, USA), and RNA integrity was determined using the Eukaryote Total RNA assay with a Nano Series II ChIP on a 2100 Bioanalyzer (Agilent, Santa Clara, CA, United States). Three independent biological replicates were pooled for RNA-seq. RNA-seq libraries were prepared using 4 µg of total RNA with the TruSeq RNA sample prep kit following the manufacturer’s protocol (Illumina). The RNA-Seq reads were aligned to the mouse GRCm38/mm10 reference genome, and expression levels for each sample were quantified and normalized using the Biomedical Genomics Workbench 5.0 (Qiagen, Venlo, The Netherlands). RNA-Seq data processing was performed with R version 4.0 (www.r-project.org, accessed on 29 April 2022).

### 2.9. Statistical Analysis

Data were presented as mean ± SD or mean ± SEM. Statistical analyses were performed in GraphPad Prism software (GraphPad, San Diego, CA, USA) using a two-tailed Student *t*-test. Differences were statistically significant when *p* < 0.05. * *p* < 0.05; ** *p* < 0.01; and *** *p* < 0.001.

## 3. Results

### 3.1. MiR-23a Is Not Required for the Embryonic Development of LCs

Recent lineage tracing studies have confirmed that adult tissue-resident macrophages (TRMs), including skin LCs, arise from embryonic precursor cells, such as yolk sac macrophages and fetal liver monocytes [26,27]. Yolk sac macrophages migrate to the fetal skin as early as embryonic day 10.5 (E10.5), while fetal liver monocytes start to colonize the developing skin from E14.5, then are recruited to epidermis around E18 [2,27]. Although most skin LC precursors originate from yolk sac macrophages at an early embryonic stage, yolk sac-derived LC precursors are largely replaced by fetal liver monocyte-derived precursors during late embryogenesis. Consequently, adult LCs are derived predominantly from fetal liver monocytes, with only a minor contribution from yolk sac-derived macrophages [2]. So far, only a few genes, including Csf1/Csf1r, PU.1, and IL34, have been reported to be required for LC precursor development during embryogenesis [2,28,29]. Given that the miR-23a-27a-24-2 cluster is a downstream target of the transcription factor PU.1, which antagonizes lymphoid cell fate acquisition [23], we predicted that miR-23a may play a role in embryonic LC development. To investigate the role of miR-23a in murine LC ontogeny and homeostasis, we first examined miR-23a expression in epidermal LC precursors at different developmental stages, including E16.5 fetal liver monocytes, E16.5 skin monocytes, E16.5 skin macrophages, E18.5 epidermal LCs, and adult epidermal LCs. miR-23a showed a moderate expression in E16.5 fetal liver monocytes, showed a trend of upregulation after migrating into skin, then sharply downregulated when differentiating into skin macrophages, followed by a sharp upregulation when recruited into epidermis at E18.5, and reached to the expression peak in adult LCs (Figure 1A). These results indicate that miR-23a is highly expressed in adult LCs and its expression is dynamically regulated during LC embryonic development, suggesting that miR-23a may play a role in embryonic LC development, postnatal differentiation, and function. 

To determine whether miR-23a is required for embryonic LC development, we generated Csf1r^iCre^miR-23a^fl/fl^ conditional knockout (cKO) mice, in which miR-23a is deficient only in Csf1r-expressing myeloid cells, including TRMs and monocytes [3,11]. Notably, miR-23a expression was robustly reduced (>80%) in LC precursors from miR-23a cKO newborns (Figure 1B). We next examined epidermal LC precursors and CD45^+^ cells in the skin from miR23-a cKO and WT littermates at postnatal day 0 (P0). However, no significant differences were identified in the frequency or number of epidermal LC precursors (CD11b^+^ F4/80^+^) and total CD45^+^ immune cells between cKO and WT mice (Figure 1C), indicating that miR-23a is not absolutely required for embryonic LC development.

### 3.2. MiR-23a Is Not Required for LC Postnatal Homeostasis and Differentiation

In mice, a single wave of early LC precursors seeds the developed epidermis between E16.5 and E18.5 [30]. This is followed by a massive burst of proliferation during the first week of postnatal life, when a typical LC network is formed, and cells begin to express CD11c and langerin (CD207) in a time-dependent manner [31]. Given that adult epidermal LCs expressed a relatively higher level of miR-23a than fetal LC precursors (Figure 1A), and miR-23a is a target of both PU.1 and TGF-b [22,23], we hypothesized that miR-23a might be involved in postnatal LC maintenance and function. To test this hypothesis, we evaluated the frequency and number of LCs in adult epidermis from miR-23a cKO and WT littermate mice via flow cytometry. As shown in Figure 2, the frequency and number of LCs and CD45^+^ cells in miR-23a cKO mice were comparable to WT mice (Figure 2A). In addition, CD207 expression in gated LCs remained comparable between miR-23a cKO and WT mice (Figure 2A). These results indicate that miR-23a is not required in LC postnatal homeostasis and differentiation.

### 3.3. MiR-23a Negatively Regulate LC Antigen Uptake but Not LC Maturation

Residing at the outermost skin layer, LCs efficiently ingest and process environmental and self-antigens, which is vital to the induction of immune defense and tolerance. To characterize the role of miR-23a in LC phagocytic capacity, freshly isolated epidermal cells from miR-23a cKO and WT mice were incubated with FITC-dextran ex vivo for 45 min at 4 °C and 37 °C. As expected, LCs from both WT and miR-23a cKO mice had low FITC-dextran^+^ after 4 °C incubation, even though LCs from miR-23a cKO showed a statistically non-significant trend of enhanced FITC-dextran uptake (Figure 2B). Interestingly, the frequency of FITC-dextran^+^ LCs was significantly higher in miR-23a cKO mice than in WT mice when cells were incubated at 37 °C (Figure 2B). Taken together, these results indicate that miR-23a plays an inhibitory role in the phagocytotic capacity of epidermal LCs.

In steady-state conditions, epidermis-resident LCs are immature and barely express co-stimulatory molecules (e.g., CD80 and CD86). However, upon exposure to inflammatory stimuli or during ex vivo culture, LCs undergo maturation by upregulating the expression of these co-stimulatory molecules and emigrate from the skin to the draining lymph nodes, which is critical for T cell priming in response to inflammation [32]. To assess the role of miR-23a in LC maturation and function, freshly purified epidermal cells from cKO and WT mice were cultured for 72 h in vitro, and LC maturation was evaluated by comparing pre-culture and post-culture cells via flow cytometry. As expected, freshly isolated epidermal LCs from both miR-23a cKO and WT control mice expressed very low and comparable levels of CD80 and CD86 (Figure 2C). However, after in vitro culture, a high number of WT LCs expressed CD80 and CD86 (CD80 in ~40%; CD86 in ~50%), while miR-23a cKO LCs showed no difference in CD80 and CD86 expression before and after culture (Figure 2D). Furthermore, miR-23a cKO and WT LCs showed similar MHC II expression levels after in vitro culture (Figure 2D). These results indicate that miR-23a is not critically required for LC maturation and MHC II upregulation during inflammation.

### 3.4. MiR-23a Deficiency Affects UV-Induced LC Repopulation

Under inflammatory conditions, such as ultraviolet (UV) light exposure, epidermal LCs emigrate to draining lymph nodes, while BM-derived peripheral blood monocytes are recruited to the epidermis to refill the depleted niche [4,33,34]. After seeding the epidermis, monocyte-derived LC precursors undergo a process of differentiation until they are phenotypically indistinguishable from the LCs of embryonic origin [33]. Given that BM-derived LC epidermal repopulation is dependent on PU.1-RUNX3 and TGF-b-Smad2/Smad3 signaling pathways [25,35] and that both TGF-b and PU.1 promote miR-23a expression [22,23], we hypothesized that miR-23a deficiency might affect epidermal LC repopulation under inflammatory conditions. To test this hypothesis, we induced skin inflammation in miR-23a cKO and WT control mice with ultraviolet-C (UVC) irradiation and evaluated epidermal LC repopulation 21 days after UVC exposure. UVC-irradiated miR-23a cKO mice had a lower frequency of epidermal LCs and CD45^+^ immune cells (Figure 3A), indicating that the process of LC repopulation is defective after UV-induced inflammation in miR-23a–deficient mice. To assess the differentiation status of LCs that repopulated the skin, we evaluated the expression of LC differentiation markers, including CD207, EpCAM, and CD11c. As shown in Figure 3B, repopulated LCs from miR-23a cKO and WT mice expressed comparable levels of CD207, EpCAM, and CD11c. These results suggest that miR-23a is dispensable for the differentiation of LC precursors that replenish the skin after an inflammatory insult, even though quantitatively fewer LCs had repopulated skin in miR-23a cKO mice than in WT mice. Csf1rCre-mediated gene deletion may potentially impact the development of monocytic LC precursors during inflammation-induced LC repopulation. To address the potential role of monocyte development in the defective post-inflammatory skin repopulation of LCs in miR-23a cKO mice, we evaluated peripheral blood monocytes. As shown in Figure 3C, comparable frequencies and numbers of peripheral blood Ly6C^hi^ and Ly6C^low^ monocytes were identified in miR-23a cKO and WT mice. Thus, monocytic LC precursor development does not contribute to the inflammation-induced LC repopulation defect in miR-23a cKO mice.

### 3.5. Molecular Mechanisms of MiR-23a-Mediated LC Regulation

To obtain mechanistic insights into the enhanced antigen uptake of LCs that lack miR-23a, we performed bulk RNA-seq analysis for steady-state adult LCs from miR-23a cKO and WT mice. A total of 1798 differentially expressed genes (*p* < 0.05) were identified in LCs from miR-23a cKO mice relative to WT mice, including 896 upregulated and 902 downregulated genes. To pinpoint the potential target genes of miR-23a and the related molecular mechanisms of miR-23a-mediated LC regulation, we performed gene ontology (GO) analysis for the upregulated genes and identified multiple biological pathways that may be dysregulated in miR-23a cKO LCs, including the following: the external side and intrinsic component of plasma membrane; the regulation of leukocyte activation and proliferation, phagocytosis and engulfment; and the regulation of cell–cell adhesion, MAPK cascade, and extracellular matrix organization (Figure 4A). These signaling pathways are closely related to the aggravated phagocytosis and antigen uptake in LCs from miR-23a cKO mice. Using a text-mining procedure in R, we generated a frequent rank-ordered list of upregulated genes from the enriched GO terms and then determined the overlapping genes contained in the RNA-seq results and the miR-23a target gene set collection acquired in the R package (multiMiR 1.19.0 R package) [36]. We observed 11 overlapping significantly associated genes (Figure 4B). Intriguingly, among these 11 significantly associated genes, *Lrp1* [37,38,39], *Abca1* [40,41], *Map4k4* [42,43], *Vegfa* [44], and *Nf1* [45] regulate cell efferocytosis, micropinocytosis, and endocytic activities in DCs, macrophages, and tumor cells. RT-PCR were performed to validate the expression levels of selected miR23a target genes including *Nf1, Abca1, Map4k4* and *Vegfα.* All four genes exhibited upward trend in LC with miR23aKO, but only Abca1 showed statistically significance (Appendix A). These results strongly suggest the critical role of miR-23a in controlling LC antigen uptake and phagocytic function through targeting multiple regulatory molecules. Given that the phagocytosis of apoptotic cells via macrophages and DCs leads to the suppression of the inflammatory immune response [46,47] and the LC-mediated phagocytosis of apoptotic keratinocytes is essential for the anti-inflammatory role of LCs in the resolution of UVB-induced cutaneous inflammation [48], the enhanced phagocytic capacity of LCs lacking miR-23a may affect their immune regulatory function. As expected, LCs from miR-23a cKO mice exhibited downregulation in a cluster of genes related to proinflammatory activities, although none of these genes are predicted or confirmed direct targets of miR-23a (Appendix A). This result indicates that miR-23a may play a proinflammatory role in LC function through the direct regulation of LC phagocytosis and efferocytosis.

To characterize the mechanism of miR-23a-mediated LC repopulation, we collected the replenished LCs from miR-23a cKO and WT mice 3 weeks after UV irradiation and performed bulk RNA-seq analysis. A total of 3940 differentially expressed genes (*p* < 0.05) were identified in replenished LCs from miR-23a cKO mice relative to WT LCs, including 2242 upregulated and 1698 downregulated genes. Surprisingly, only 204 of the 2242 upregulated genes from replenished miR-23a cKO LCs overlapped with those genes that were upregulated in the steady-state miR-23a cKO LCs (Figure 4D), indicating a diversified regulatory role for miR-23a in LCs under steady-state versus replenished LCs under inflammatory conditions. Apoptosis and cell cycle signaling pathway gene enrichment analysis was performed to identify potential miR-23a targets involved in regulating UV-induced LC repopulation. Relative to replenished cells from WT mice, replenished miR-23a cKO LCs had a dramatic increase in the expression of the components in apoptosis signaling pathway, but not in cell cycle signaling pathways, including the proapoptotic genes *Fas/FasL*, *Bax*, *Bak*, *Noxa*, *Fas,* the downstream initiator caspase (*Caspase 8*), and effector caspases (*Caspase3*, *Caspase6*, and *Caspase7*) (Figure 4E). *Fas* activation is known to trigger apoptosis in many cell types and plays a central role in the physiological regulation of programmed cell death [49,50]. *Bax* and *Bak* belong to the Bcl-2 family of proapoptotic effector proteins, and the activation and oligomerization of *Bax* and *Bak* mediate mitochondrial outer membrane permeabilization (MOMP), which is a key step in apoptosis [51]. Bcl-2 homology (BH) domain 3-only proteins, such as *Puma*, *Noxa*, *Bid*, and *Bim*, promote cell death by inactivating antiapoptotic Bcl-2 family members and directly activating proapoptotic BH domain proteins (*Bax* and *Bak*), which ultimately leads to MOMP and the mitochondrial release of proapoptotic molecules [51]. Interestingly, luciferase reporter assays, FACS analysis, and qPCR have all confirmed that *Fas* is a direct target of miR-23a [52]. In addition, the proapoptotic BCL-2 family proteins *Noxa* (*Pmaip1*), *Puma* (*Bbc3*), and *Bax* have been shown to be the direct targets of miR-23a, and the downregulation of miR-23a plays an important role in traumatic brain injury- and irradiation-induced neuronal apoptosis [53,54]. These results indicate a context-dependent regulatory role for miR-23a in maintaining LC homeostasis at least through targeting Fas and Bcl-2 family proapoptotic proteins and related cell death pathways. To further substantiate the involvement of apoptotic signaling molecules and pathways in miR-23a-mediated LC repopulation, we evaluated the apoptosis of repopulated LC, 3 weeks post-UV treatment, in miR23aKO mice compared to littermate controls. As shown in Figure 4F, LC from miR23aKO mice exhibit a significant increase in apoptosis when compared to their littermate controls. This result serves as a biological validation of miR-23a’s regulatory role in LC repopulation homeostasis, accomplished through the targeting of pro-apoptotic molecules and pathways.

## 4. Discussion

A key role for TRMs is the maintenance of immune homeostasis, which is achieved partly through their scavenger function, where they quickly clear debris from dying cells under steady-state conditions and after pathological conditions, such as infection. In the epidermis, LCs constantly extend and retract dendrites between keratinocytes and use a similar innate mechanism in surveillance and control of local epidermal immune tolerance [48,55]. In LCs, TGF-b-induced AXL expression promotes the uptake of GAS6-expressing apoptotic keratinocytes and inhibits inflammatory cytokine production [56]. Consistent with the enhanced antigen uptake and phagocytic function in LCs with miR-23a deletion, the GO term enrichment analysis of RNA-seq data from these cells showed a significant enrichment in gene families, such as the components of plasma membrane, phagocytosis, engulfment and recognition, the positive regulation of leukocyte activation, the positive regulation of cell–cell adhesion, MAPK cascade, and antigen processing and presentation. In support of this analysis, we identified several genes that are direct targets of miR-23a within a cluster of GO term-related genes that were upregulated in LCs lacking miR-23a. The low-density lipoprotein receptor-related protein-1 (Lrp-1) is a large multifunctional endocytic receptor belonging to the low-density lipoprotein receptor family [39], and Lrp-1 is one of the validated miR-23a targets [57]. In addition to regulating cholesterol endocytic transport [58,59], Lrp1 has been shown to be a macrophage efferocytosis receptor and a contributor to DC efferocytosis, cross presentation, and immune activation [37,38,59], including the increased uptake of apoptotic cells in glucocorticoid stimulated macrophages. The ATP binding cassette transporter A1 (*Abca1*) is a membrane transporter that is abundantly expressed in macrophages, sometimes inhibited by Lrp1, and causes cholesterol efflux in human macrophages [60]. In addition to mediating cholesterol efflux, *Abca1* also promotes macrophage efferocytosis by regulating the release of “find-me” ligands [40] and is associated with cell phagocytosis, clearance, and anti-inflammatory reactions through promoting the secretion of annexin A1 [41]. A member of the serine/threonine protein kinase family, mitogen-activated protein 4 kinase 4 (*Map4k4*), functions upstream of MAPK cascades, controlling a large number of distinct cellular processes [61], including cholesterol deposition, and it has been experimentally validated as an miR-23a target [62]. In medulloblastoma tumor cells, *Map4k4* enhances endocytic activity and the depletion of Map4k4 impairs hepatocyte growth factor-induced dextran endocytosis [43]. Vascular endothelial growth factor A (Vegfa) is a multifunctional cytokine that mediates the growth of new blood vessels by binding to the cell surface receptors Vegfr1 and Vegfr2, which are expressed on vascular endothelial cells. Interestingly, it has been shown that the uptake of apoptotic cells by murine alveolar macrophages and human monocyte-derived macrophages is inhibited by Vegf depletion, while Vegf overexpression significantly enhances apoptotic cell uptake by alveolar macrophages in vivo, indicating a positive regulatory role for Vegf in macrophage endocytosis and clearance of apoptotic cells [44]. Furthermore, Vegf-C/Vegfr-3 signaling increases the macrophage efferocytosis of apoptotic neutrophils via the upregulation of macrophage integrin αv [63], further indicating a role for Vegf in regulating macrophage phagocytosis.

In steady-state conditions, fully differentiated LCs maintain their numbers in the skin through a low rate of in situ proliferation, and the LCs self-renew without further input from the BM; however, under inflammatory conditions such as UV light exposure, hematopoietic precursors reconstitute the LC pool in the inflamed skin [5]. Although previous studies provide a clear consensus on the role of adult monocytes in maintaining and replenishing tissue macrophages in other organs, the nature of precursors that repopulate LCs in the epidermis has remained controversial. Using a murine model of UV irradiation–induced acute skin inflammation, two studies have produced conflicting results on whether Gr-1^hi^ monocytes persist as eventual LC precursors in the epidermis of UV-treated mice [4,37]. A recent study using a BM transfer plus LC deletion mouse model, showed the sequential differentiation pathway of monocyte-derived LC precursors in the epidermis, which ultimately acquired a similar capacity for self-renewal as steady-state epidermal LCs [33]. This result suggests that the so-called “short-term CD207^low^”- and “long-term CD207^hi^”-replenished LCs [4] may represent different differentiation stages of skin repopulating LCs derived from blood monocytes.

The transcription factor PU.1 promotes myeloid differentiation by upregulating the miR-23a-27a-24-2 cluster, which is involved in promoting myeloid development but antagonizing lymphoid cell fate acquisition in hematopoietic progenitors [16,23]. This suggests a potential connection between the miR-23a-27a24-2 cluster’s myeloid promoting effect and the UV-induced LC skin repopulation defect we observed in miR-23a cKO mice. However, in vitro and in vivo mechanistic studies indicate that miR24-2, but not miR-23a and miR-27a, is the dominant player in the myeloid promoting effect through the direct targeting of Tribbles homolog 3 gene expression [16]. This notion is further supported by the comparable monocyte distributions seen in both the BM and peripheral blood of miR-23a cKO and WT mice (Figure 3C).

The cell surface receptor *Fas* belongs to the tumor necrosis factor receptor family subgroup. *Fas* triggers apoptosis and plays a critical role in controlling immune system homeostasis and tumor suppression [64,65]. The activation of *Fas* via its physiological ligand FasL causes the assembly of an intracellular death-inducing signaling complex (DISC) containing FADD and the cysteine protease caspase-8, which activates downstream effector caspases, such as caspase-3 and caspase-7, leading to the cleavage of vital cellular proteins [66]. In addition to the *Fas*-mediated extrinsic apoptosis pathway, there are two Bcl-2 homology (BH) domain-containing proapoptotic subfamilies that facilitate cell death by promoting mitochondrial outer membrane permeabilization (MOMP). BH3-only subfamily proteins, including Bid, Bad, Bim, and Bik, along with the multidomain proteins Bax and Bak, are required for MOMP and the activation of the caspase cascade in the Bcl-2 regulated (or intrinsic) apoptosis signaling pathway; on the other hand, pro-survival Bcl-2 family proteins, including Bcl-2, Bcl-XL, Bcl-W, and A1, function to keep the proapoptotic Bcl-2 family proteins in check and are essential for cell survival [67]. The upregulation of multiple signaling molecules within both the extrinsic and intrinsic apoptosis signaling pathways in the miR-23a deletion LCs that had replenished the skin after UV exposure indicates that enhanced cell apoptosis is involved in the defect we observed in LC epidermal repopulation after UV treatment (Figure 4E). Intriguingly, both Fas and Bax, the key molecules in the extrinsic and intrinsic apoptosis signaling pathways, respectively, have previously been confirmed as direct targets of miR-23a [52,53,54], which further suggests a critical role for miR-23a in controlling homeostasis in repopulated epidermal LCs through targeting multiple apoptosis pathways. Interestingly, transcriptome analysis showed that steady-state LCs and repopulated LCs after UV exposure, lacking miR-23a, shared very few (7%) upregulated genes. (Figure 4D). Notably, apoptotic signaling pathway molecules were upregulated only in repopulated LCs after UV exposure, but not in steady-state LCs, lacking miR-23a (Figure 4E). These data indicate different regulatory roles for miR-23a between embryonic-derived steady-state LCs and inflammation-induced monocyte-derived LCs that replenish the skin after inflammation. In a study that used BM transfer plus an immune injury-induced LC deletion mouse model, although overall transcriptomes were virtually indistinguishable between steady-state LCs and replenished LCs 10 weeks after transfer, the expression of a cluster of genes associated with cell adhesion, mobility, and survival was dramatically different [33]. These findings are somewhat different from what we observed, in which the experimental setting, the dynamics of LC replenishment and the different time points after inflammation at which the LCs were tested may be factors that affect LC differentiation and accommodation after monocytic precursors have seeded the epidermis.

Previous studies have shown that miR-23a plays various roles in the initiation, progression, and treatment of human cancers. Downregulated miR-23a was often observed in myeloid leukemia and melanoma [21]. Our current results on the roles of miR-23a in skin LC phagocytosis function and inflammation-induced repopulation suggest a potential alternative way for miR-23a in regulating human myeloid or skin cancer development via immune regulation, which is supported by the role of miR-23a in regulating tumor associated macrophage differentiation [68]. Further investigation is required to explore the specific roles of miR-23a in LC antigen processing and presentation, its involvement in regulating skin inflammation, as well as its impact on human LC function and repopulation.

## 5. Conclusions

Overall, our results indicate the diverse regulatory roles for miR-23a in steady-state LCs and inflammation-induced LC repopulation, representing an extra epigenetic regulatory layer that incorporates TGF-b- and PU.1-mediated LC regulation (Figure 5). In steady-state LCs, miR-23a inhibits phagocytosis and antigen uptake by targeting multiple regulatory molecules related to efferocytosis and endocytosis, which may represent a feedback mechanism for controlling LC phagocytic and immune regulatory function. Nonetheless, in freshly replenished LCs after an inflammatory insult, miR-23a promotes LC homeostasis by targeting multiple proapoptotic regulators.

## Figures and Tables

**Figure 1 biology-12-00925-f001:**
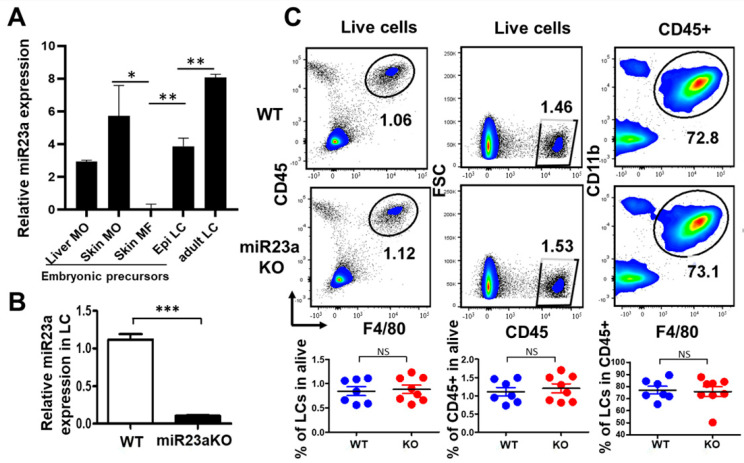
miR-23a is not required for the embryonic development of LCs. (**A**) miR-23a expression was evaluated with miRNA array for different embryonic LC precursors, including E16.5 fetal liver monocytes (Liver MO), E16.5 skin monocytes (skin MO), E16.5 skin macrophages (skin MF), E18.5 epidermal LCs, and adult steady-state LCs (n = 3). (**B**) P0 epidermal LC precursor expression of miR-23a by RT-PCR in Csf1icremiR-23afl/fl (miR-23a cKO) and WT littermates (n = 3). (**C**) LC precursor frequency and numbers from P0 pups were evaluated in miR-23a cKO and WT mice. Each dot represents one newborn pup (n = 7–8) and bars represent mean ± SD. For statistical analysis, Student’s two-tailed unpaired *t*-test was applied. NS, not significant; * *p* < 0.05, ** *p* < 0.01, *** *p* < 0.001.

**Figure 2 biology-12-00925-f002:**
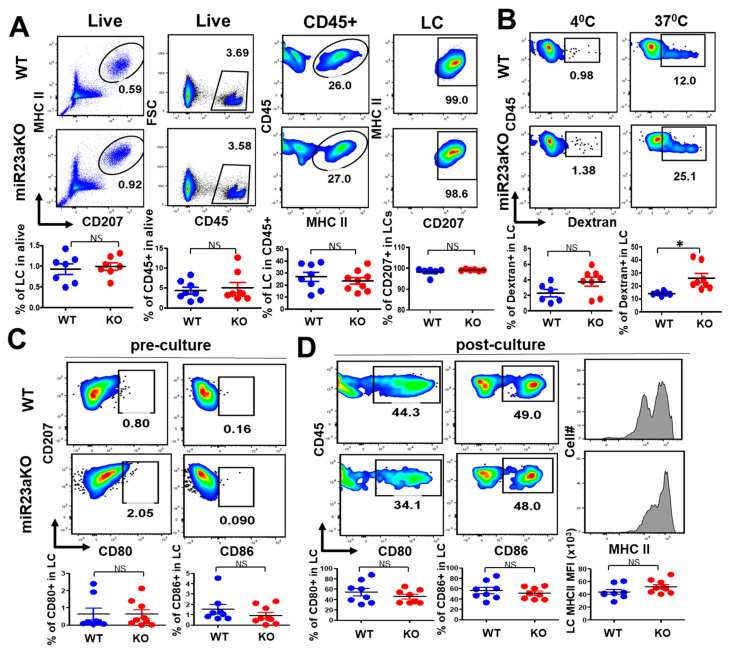
**miR-23a controls LC antigen uptake but is dispensable for LC homeostasis and maturation.** (**A**) Adult epidermal CD45+ LC frequencies were evaluated via flow cytometry for miR-23a cKO and WT mice (n = 8–9). (**B**) Epidermal cells were incubated with 0.025% dextran-FITC for 45 min at 37 °C or 4 °C, then stained with anti-CD45 and anti-MHC-II and analyzed using flow cytometry (n = 6–8). (**C**) Freshly prepared epidermal cell suspensions were stained with anti-MHC-II, -CD45, -CD80, and -CD86 antibodies and analyzed via flow cytometry (n = 8). (**D**) Epidermal cell suspensions were cultured in complete culture medium for 72 hrs and then stained and analyzed as described in (**C**) (n = 8). Bars represent mean ± SD from 2–3 independent experiments. For statistical analysis, Student’s two-tailed unpaired *t*-test was applied. NS, not significant; * *p* < 0.05.

**Figure 3 biology-12-00925-f003:**
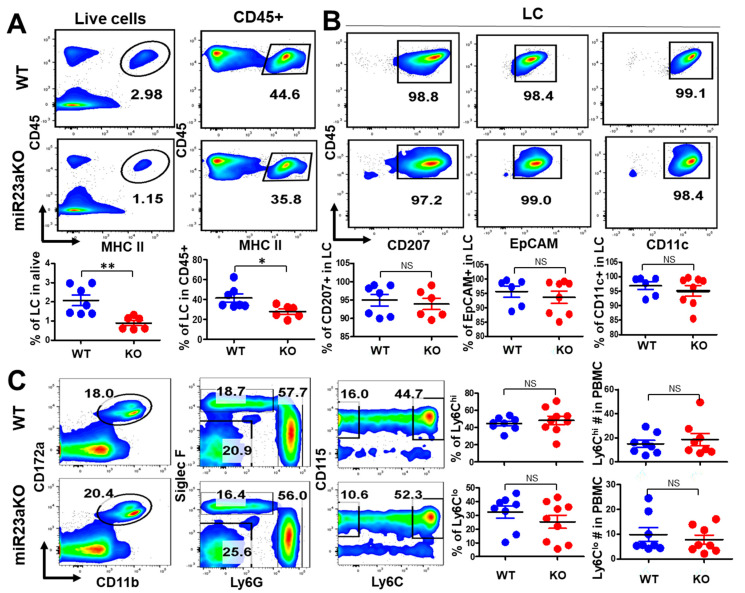
**miR-23a is required for inflammation-induced LC repopulation.** Mice were first exposed to ultraviolet (UV) irradiation for 15 min. Three weeks after UV exposure, epidermal LC frequency and number (**A**) and LC differentiation status (**B**) were evaluated via flow cytometry for miR-23a cKO and WT mice (n = 6–9). (**C**) The frequency of peripheral blood LY6C^hi^ and LY6C^low^ monocytes was evaluated using flow cytometry in miR-23a cKO and WT mice (n = 8–9). Bars represent mean ± SD from 2–3 independent experiments. For statistical analysis, Student’s two-tailed unpaired *t*-test was applied. * *p* < 0.05, ** *p* < 0.01.

**Figure 4 biology-12-00925-f004:**
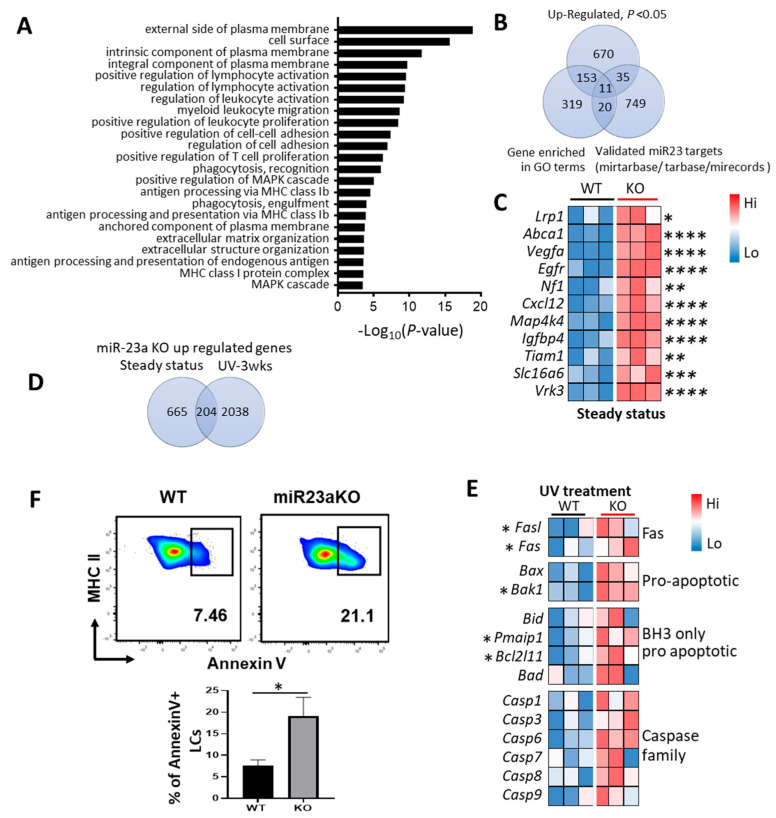
**Molecular mechanisms of miR-23a-mediated regulation of LC antigen uptake and inflammation**−**induced repopulation.** Epidermal LCs from steady-state conditions and 3 weeks after UV exposure were sorted from miR-23a cKO and WT mice. Bulk RNA sequencing was performed on the sorted cells. (**A**) Gene ontology (GO) biological process enrichment analysis on the upregulated genes in steady-state miR-23a cKO LCs. Top significantly enriched GO (−log_10_(*p*-value)) terms were selected and displayed. (**B**) Venn diagram of differentially expressed genes in steady-state miR-23a cKO LCs, GO term-enriched genes, and validated miR-23a targets in the databases. (**C**) Heatmap showing the paired expression of the overlapping 11 miR-23a target genes (**B**) in WT and miR-23a cKO LCs at steady state. (**D**) Venn diagram of differentially expressed genes during steady state and 3 weeks after UV exposure for miR-23a cKO LCs. (**E**) Heatmap showing the selected proapoptotic gene expression in WT and miR-23a cKO LCs 3 weeks after UV exposure. (**F**) Apoptosis of repopulated LC after UV irradiation was evaluated via annexin V staining using flow cytometry. T * *p* < 0.05; ** *p* < 0.01; *** *p* < 0.001; **** *p* < 0.0001.

**Figure 5 biology-12-00925-f005:**
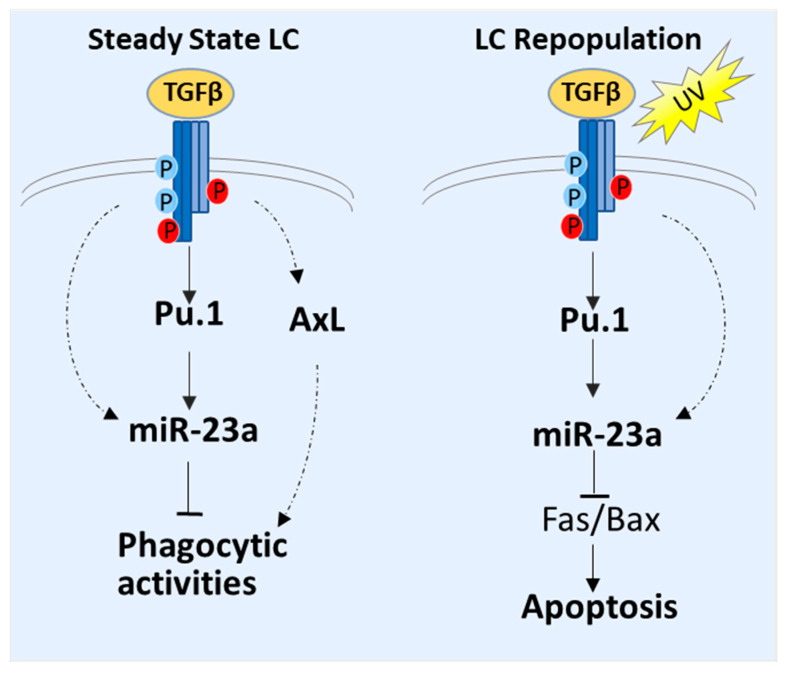
The speculated schematic model of miR-23a in the regulation of LC phagocytic activities in steady state and LC skin repopulation after UV exposure.

## Data Availability

All data that support the findings of this study are available from the corresponding author upon reasonable request.

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
