# Peer review of "MiR-23a Regulates Skin Langerhans Cell Phagocytosis and Inflammation-Induced Langerhans Cell Repopulation"

_biology, 2023, doi:10.3390/biology12070925_

Round 1

Reviewer 1 Report

The authors present a compelling series of experiments regarding the role of miR-23a in murine Langerhans cells. The results are clearly and logically laid out, and explained in great depth in relation already existing data in the field. 

I have three (very) minor recommendations to the authors regarding their manuscript:

1. In the introduction LCs are twice labelled dendritic cells, but then later are correctly called tissue resident macrophages. I would recommend to call the cells "resident macrophages that act as dendritic cells", or something similar. Even though these terms might be clunkier, they are more precise.

2. The quality of the figures is low in the word format, and even in the supplementary file they are quite pixelated. I recommend higher dpi settings on whatever program was used to generate the images. I also don't quite understand the reason that some flow cytometry dot plots are in black and white while others use pseudocolor. I recommend pseudocolor for all such dot plots.

3. There is very little mention of human data in relation to murine findings in the discussion. While I appreciate that there might be limited human data to reference, this should be mentioned in the discussion.

Reviewer 2 Report

The manuscript is fine and okay for the publication. Following points should be addressed.

1. Number of mice were included the research missing.

2. Discussion part should be precise and according to the results.

3. Future direction after the objective of the study achieved is missing.

4. Conclusion needs refinements.

Reviewer 3 Report

Dear authors,

The paper, addressing the role of miR-23a in regulation of skin Langershans cell phagocytosis and in inflammation-induced repopulation, is based on studies performed in LC cells isolated from wt and conditional miR-23a ko mice. Although the authors demonstrate no role for miR-23a in embryonic LC development or postnatal homeostasis, they have shown an important role for miR-23a in LC antigen uptake and inflammation-induced LC epidermal repopulation.

1 - Importantly one RNA-seq analysis performed on adult LCs from miR-23a cKO and WT mice have revealed 11 genes with miR-23a targeting sites within the over expressed genes in miR23a cKO LCs. As these genes are involved in endocytic activities in DCs among others, the authors propose that miR-23a are controlling LC antigen uptake and phagocytic function through targeting these multiple regulatory molecules. However, no further characterisation of these potential target genes is made, namely before and after inflammatory challenges in wt mice or in the comparison between wt and ko mice upon challenge. As such, the expression levels of these target genes should be characterised in order to determine whether there is a negative correlation with their miRNA expression levels in order to validate the RNA-seq data obtained. Furthermore, luciferase assays should be performed for the most functionally relevant genes in order to determine if they constitute direct miR-23a targets.

2 - In addition, the authors have demonstrated an inflammation-induced LC epidermal repopulation and here again, they have characterised by RNA-seq LCs collected from miR-23a cKO and WT mice 3 weeks after UV irradiation, finding some overexpressed genes, most of which do not overlap with those of the steady state. These over expressed genes are mostly related to the apoptotic pathway but not with the cell cycle functions, some of which - Fas and Bcl-2 family members - had previously been confirmed to be direct targets of miR-23a. Again, in the present study, further characterisation of these target genes should be performed in samples depleted from miR-23a or in LC cells subject to challenging conditions. These control experiments are missing from the system.

3 - In addition to the experiments suggested above it would be important to understand wether these target genes change in embryonic development in inverted correlated levels to those of miR-23a /as shown in figure 1A. Would these genes be targeted by miR-23a in development or only in adult LCs subject to challenges? The authors could discuss this issue. 

Round 2

Reviewer 3 Report

I thank the authors for their replies as they have addressed my questions and make the paper clearer for the readers.